# Effects of Two Compound Probiotic Formulations on Gastrointestinal Symptoms and Gut Microbiota: A 4-Week Randomized, Double-Blind Intervention Trial

**DOI:** 10.3390/nu17172886

**Published:** 2025-09-06

**Authors:** Zhen Qu, Ying Wu, Yiru Jiang, Jiajia Fan, Li Cao, Yao Dong, Shuguang Fang, Shaobin Gu

**Affiliations:** 1College of Food and Bioengineering, Henan University of Science and Technology, Luoyang 471000, China; q819707110@163.com (Z.Q.); wuying2000@126.com (Y.W.); 17719253702@163.com (Y.J.); 13290501131@163.com (J.F.); xbcaoli@163.com (L.C.); 2Henan Engineering Research Center of Food Material, Henan University of Science and Technology, Luoyang 471023, China; 3Wecare Probiotics R&D Centers (WPC), Wecare Probiotics Co., Ltd., Suzhou 215200, China; yao.dong@wecare-bio.com; 4Henan Engineering Research Center of Food Microbiology, Luoyang 471000, China

**Keywords:** multi-strain probiotic formulation, gastrointestinal discomfort symptoms, intestinal flora, randomized double-blind study, dose-response, dose-ranging

## Abstract

**Background/Objectives**: Probiotic interventions can alleviate gastrointestinal (GI) discomfort, but evidence comparing multi-strain combinations at different doses remains limited. We evaluated whether formulation potency influences clinical and microbiome outcomes. **Methods**: In a 4-week, randomized, double-blind trial, 100 eligible adults received one of two higher-dose multi-strain probiotic formulations at different dosages (Wec120B vs Wec300B). Weekly Gastrointestinal Symptom Rating Scale (GSRS) scores tracked symptom trajectories. Gut microbiota composition and diversity were profiled by 16S rRNA gene sequencing. Biomarkers included lipopolysaccharide (LPS), fecal calprotectin (FC), and immunoglobulin A (IgA). **Results**: Results indicated that the Wec120B group showed more significant improvement in abdominal pain during the early phase of intervention, while the Wec300B group was more effective in relieving reflux symptoms. In terms of biomarkers, Wec120B was more effective in reducing lipopolysaccharide (LPS) levels, whereas Wec300B showed a greater increase in immunoglobulin A (IgA) and a more pronounced reduction in fecal calprotectin (FC) levels. Both formulations significantly increased the abundance of beneficial genera such as *Bifidobacterium*, *Blautia*, *[Eubacterium]_hallii_group*, and *Anaerostipes*, while suppressing the growth of potential pathogens including *Prevotella* and *Escherichia-Shigella*. **Conclusions**: These findings suggest that both compound probiotic products can significantly improve GI symptoms and modulate gut microbiota structure, with Wec300B showing a superior performance in microbial regulation, likely due to its higher dosage of probiotics. This study provides reference evidence for the rational application of probiotic products in gut health management.

## 1. Introduction

With the continued advancement of the “Healthy China 2030” initiative, gut health has emerged as a critical foundation for population-wide well-being. According to a 2021 global survey conducted by the Rome Foundation, approximately 40.3% of individuals met the diagnostic criteria for functional gastrointestinal disorders (FGIDs), and the prevalence of functional dyspepsia in China has reached 23.5%. Regional studies have reported similar trends, such as a 32.0% overall prevalence of FGIDs in Hainan Province, with 10.7% specifically attributed to functional dyspepsia [1]. Gastrointestinal dysfunction not only significantly reduces quality of life but is also closely associated with increased risks of depression [2] and metabolic syndrome [3]. In 2018, the United States spent an estimated USD 119.6 billion on GI-related healthcare, underlining the substantial economic burden associated with these conditions [4].

In the post-COVID-19 era, antibiotic overuse has further disrupted gut microbiota composition. Data from the U.S. Centers for Disease Control and Prevention (CDC) in 2022 revealed that over 236 million outpatient antibiotic prescriptions were issued, with at least 28% considered unnecessary [5]. In response, the global probiotics market has seen an explosive growth. The global probiotics market is poised for substantial growth. Forecasts indicate that the market will surge, expected to reach a staggering $220.14 billion by 2030. This exponential growth represents a compound annual growth rate (CAGR) of 14.0% from 2023 to 2030 [6].

However, the 2023 CCTV “3·15” Consumer Rights Day program exposed several cases of false advertising related to probiotic products, highlighting the reality that the fundamental research in this field lags behind rapid industrial development. In particular, regarding the synergistic effects of multi-strain probiotic formulations and their dose–response relationships, no scientific consensus has yet been established [7]. The influence of different strain combinations and dosages on clinical outcomes remains complex and variable, warranting further in-depth studies to elucidate the underlying mechanisms and to define standardized guidelines.

The gut microbiota has been vividly referred to as the host’s “second genome” [8], with its extensive genomic repertoire and diverse metabolites exerting profound effects on the host’s physiological functions [9]. Among these metabolites, short-chain fatty acids (SCFAs) not only serve as an energy source for intestinal epithelial cells but also maintain mechanical, chemical, microbial, and immune barriers of the gut, thereby regulating immune homeostasis and the balance of digestion and absorption [10]. In addition, gut microbes utilize bile acid metabolic pathways to generate secondary bile acids, which act through bile acid receptors to participate in immune modulation and metabolic regulation [11]. More importantly, microbially derived SCFAs, secondary bile acids, and indole derivatives can activate enteroendocrine cells, promoting the secretion of glucagon-like peptide-1 (GLP-1) and thereby influencing host neuroendocrine regulation and energy metabolism [12]. Notably, an imbalanced Firmicutes-to-Bacteroidetes ratio has been linked to an increased severity of irritable bowel syndrome (IBS) symptoms [13]. Several studies have demonstrated the beneficial effects of probiotics in alleviating IBS and modulating immune responses [14]. For instance, Lactobacillus rhamnosus GG has been shown to enhance mucin (MUC2) expression and improve gut barrier integrity [15], while multi-strain probiotics may provide broader metabolic benefits, such as increasing SCFA production through complementary fermentation pathways [16].

Nevertheless, key challenges remain. Firstly, the dose–response relationship remains poorly defined, with most trials using fixed doses and lacking gradient comparisons. Secondly, the mechanisms underlying inter-strain synergy are largely unexplored, as many clinical studies prioritize symptomatic outcomes over mechanistic insights into microbial–host metabolic interactions. Finally, objective biomarkers for probiotic efficacy remain underdeveloped.

To address these gaps, we evaluated two multi-strain probiotic formulations delivered as 2-g sachets that differed in dosage, 120 billion (Wec120B) vs 300 billion (Wec300B) colony-forming units (CFU) per dose. In a 4-week randomized, double-blind, controlled trial in adults with GI discomfort, we assessed effects on gastrointestinal symptoms and characterized changes in gut microbiota composition, immune biomarkers, and gut-barrier indicators.

## 2. Materials and Methods

### 2.1. Study Design

This was a randomized, double-blind, placebo-controlled clinical trial conducted between December 2024 and January 2025 at the School of Food and Biological Engineering, Henan University of Science and Technology. This study followed the principles of the Declaration of Helsinki [17] and was approved by the Medical Ethics Review Committee of Henan University of Science and Technology (Approval No. 2024-002, 13 December 2024). The trial was registered at ClinicalTrials.gov (NCT07025798).

Participants were recruited voluntarily through community advertisements and outpatient clinics. All participants were screened for eligibility based on predefined inclusion and exclusion criteria. After screening, eligible participants were randomly assigned (1:1) to one of two intervention groups. The intervention lasted for 4 weeks, during which participants consumed the assigned probiotic product daily. They completed weekly follow-up questionnaires (weeks 0, 1, 2, 3, and 4) and provided fecal samples at weeks 0, 2, and 4.

A CONSORT flow diagram (Figure 1) illustrates the participant recruitment, randomization, follow-up, and analysis. To ensure safety and compliance, participants were instructed to record daily intake and any adverse events in a diary card. Study personnel monitored adherence through sachet counts and the recovery of unused products at the end of the study. Written informed consent was obtained from all participants prior to enrollment.

### 2.2. Inclusion and Exclusion Criteria

Inclusion criteria: (1) voluntary participation with written informed consent; (2) ability to comply with the study protocol; (3) aged between 18 and 65 years; (4) diagnosed with gastrointestinal dysfunction according to the “Chinese Consensus on Precision Health Communication—Public Gut Health Guidelines”; (5) symptoms including irregular bowel movements, lose or hard stools, borborygmi, bloating, belching, excessive flatulence, abdominal pain, acid reflux, heartburn, hunger-induced stomach or abdominal pain, and nausea; (6) clinical signs such as abdominal pain or bloating, diarrhea or constipation, dyspepsia, acid regurgitation, halitosis or foul-smelling gas, skin issues, and abnormal stool color or shape.

Exclusion criteria: (1) use of medications affecting gut microbiota (e.g., antibiotics, probiotics, intestinal mucosal protectants, traditional Chinese medicines) for more than one week within one month prior to screening; (2) recent intake of substances with similar functions that may interfere with the outcomes; (3) use of antibiotics during illness; (4) presence of severe systemic disease or malignancy; (5) pregnancy, lactation, or plans for conception during the study; (6) inability to participate due to personal reasons; (7) deemed unsuitable for participation by the investigators.

### 2.3. Sample Size and Randomization

The sample size was estimated based on previous studies investigating the effects of probiotics on gastrointestinal symptoms, assuming an effect size of 0.6 for GSRS score reduction, α = 0.05, and 80% power. The calculation indicated that at least 45 participants were required per group. Considering a potential dropout rate of 10–20%, a total of 106 participants were initially recruited. Eligibility was assessed according to predefined inclusion and exclusion criteria, and all participants were provided with a blank informed consent form and a detailed explanation of the study. After excluding 6 individuals, 100 participants (50 in each group) were finally enrolled.

Randomization was performed using a computer-generated random number sequence (SPSS randomization module). Allocation concealment was ensured by using sequentially numbered, opaque, sealed envelopes prepared by an independent statistician not involved in study procedures. Both participants and investigators were blinded to group assignments until data analysis was completed.

### 2.4. Participants and Intervention

A total of 100 participants diagnosed with functional gastrointestinal disorders were screened and randomly assigned to either the Wec120B group or the Wec300B group (*n* = 50 per group). Participants in the Wec120B group consumed two sachets of Wec120B daily (2 g/sachet, release-tested viable count ≥ 1.2 × 10^11^ CFU/sachet). This formulation consisted of nine probiotic strains: *Bifidobacterium animalis* subsp. *lactis BLa80*, *Weizmannia coagulans BC99*, *Lacticaseibacillus rhamnosus LRa05*, *Lactiplantibacillus plantarum Lp05*, *Pediococcus acidilactici PA53*, *Bifidobacterium longum* subsp. *longum BL21*, *Bifidobacterium breve BBr60*, *Lacticaseibacillus paracasei LC86*, and *Lactobacillus acidophilus LA85*. Participants in the Wec300B group consumed two sachets of Wec300B daily (2 g/sachet, release-tested viable count ≥ 3 × 10^11^ CFU/sachet). This formulation consisted of nine probiotic strains: *Bifidobacterium animalis* subsp. *lactis BLa80*, *Weizmannia coagulans BC99*, *Lacticaseibacillus rhamnosus LRa05*, *Lactiplantibacillus plantarum Lp05*, *Pediococcus acidilactici PA53*, *Bifidobacterium longum subsp. longum BL21*, *Bifidobacterium breve BBr60*, *Lacticaseibacillus paracasei LC86*, and *Lactobacillus acidophilus LA85*. The intervention lasted for 4 weeks. Questionnaires were collected at weeks 0, 1, 2, 3, and 4, and fecal samples were obtained at weeks 0, 2, and 4. All remaining products and packaging were recovered after the study.

In the present study, two probiotic formulations with similar strain compositions but different viable counts were intentionally selected. This design was based on the study objective of examining whether the level of probiotic activity influences clinical outcomes and gut microbiota modulation in individuals with gastrointestinal discomfort. By maintaining comparable strain diversity while varying CFU counts, we aimed to isolate the dose–response relationship of probiotics, thereby reducing confounding factors related to differences in strain combinations. Such a strategy not only reflects real-world conditions in which probiotic products are frequently marketed at different doses, but also provides valuable insights into whether higher probiotic activity confers incremental benefits beyond those achievable with a standard-dose formulation.

### 2.5. Primary and Secondary Outcomes

The primary outcome was the improvement of gastrointestinal function assessed weekly using the Gastrointestinal Symptom Rating Scale (GSRS). Secondary outcomes included changes in gut microbiota composition and diversity, and fecal levels of immune markers (IgA, FL), inflammatory markers (FC, NGAL), and gut barrier indicators (LPS, D-LA, DAO).

### 2.6. Gastrointestinal Symptom Assessment

The GSRS questionnaire was administered weekly via the Wenjuanxing platform (a validated online survey system in China). The GSRS evaluates the severity of common GI symptoms on a Likert scale (1 = no discomfort, 7 = very severe discomfort), covering five domains: reflux, abdominal pain, indigestion, diarrhea, and constipation.

### 2.7. Fecal Sample Collection and Biomarker Analysis

Fresh fecal samples were collected after an overnight fast of at least 10 h and stored at −80 °C under sterile conditions. Fecal immune (IgA, FL), inflammatory (FC, NGAL), and gut barrier (LPS, D-LA, DAO) biomarkers were quantified using commercial enzyme-linked immunosorbent assay (ELISA) kits (Hepeng (Shanghai) Biotechnology Company, China; Catalog number: IgA HP-E11520, FL HP-E11678, FC HP-E10812, NGAL HP-E10253, LPS HP-E10787, D-LA HP-E10918, DAO HP-E10940), following the manufacturers’ protocols.

### 2.8. Gut Microbiota Analysis

Fecal samples from the Wec120B and Wec300B groups were analyzed using 16S rDNA sequencing to profile gut microbial communities. The sequencing workflow included genomic DNA extraction and quality assessment, PCR amplification of the target region, the purification of amplification products, library construction, and quality control, followed by high-throughput sequencing. Raw paired-end reads were merged based on overlapping regions, and low-quality sequences as well as chimeric reads were filtered to generate high-quality clean data. Subsequently, denoising procedures were applied to remove PCR and sequencing errors, yielding representative amplicon sequence variants (ASVs) and corresponding abundance tables. The resulting ASV data were subjected to analyses of α-diversity, β-diversity, taxonomic composition, and differential taxa, as well as functional composition and differential functional profiles.

### 2.9. Statistical Analysis

All statistical analyses were performed using GraphPad Prism 10.0 and SPSS 22.0 software [18]. Quantitative data were expressed as mean ± standard deviation (SD). Between-group and within-group comparisons were conducted using *t*-tests. A *p*-value < 0.05 was considered statistically significant.

## 3. Results

### 3.1. Baseline Characteristics

All 100 participants completed the 4-week intervention. In the Wec120B group, there were 29 males and 21 females with an average age of 21.1 ± 5.18 years. The Wec300B group included 24 males and 26 females, with an average age of 20.42 ± 2.7 years. The participant demographic details are summarized in Table 1.

### 3.2. Effects on Immune, Inflammatory, and Gut Barrier Markers

The post-intervention changes in fecal immune markers are shown (Figure 2A,B). Both groups demonstrated a significant increase in fecal IgA and a decrease in fecal lactoferrin (FL) levels. The Wec300B group exhibited a more pronounced IgA elevation (from 1249.63 μg/mL to 1552.87 μg/mL). IgA is a key mucosal antibody involved in the immune exclusion of pathogens and microbiota homeostasis [19], while elevated FL is indicative of neutrophil-driven intestinal inflammation [20].

As shown (Figure 2C,D), the levels of fecal calprotectin (FC) and neutrophil gelatinase-associated lipocalin (NGAL) decreased significantly in both groups, with a stronger FC reduction in the Wec300B group (from 696.94 ng/mL to 477.56 ng/mL). FC is a stable neutrophil protein and a reliable marker of intestinal inflammation [21]. NGAL is involved in barrier integrity and microbial regulation, making it a valuable indicator of gut mucosal health [22].

The figures illustrate the modulation of gut barrier markers including LPS, D-lactate (D-LA), and diamine oxidase (DAO) (Figure 2E,F). All three markers decreased significantly after 4 weeks, with the Wec120B group showing a greater reduction in LPS (from 375.72 EU/L to 277.72 EU/L). Elevated LPS reflects increased intestinal permeability and endotoxemia risk [23], while D-LA and DAO are indicators of compromised mucosal integrity [24].

### 3.3. Comparison of Gastrointestinal Symptoms Before and After Intervention

Gastrointestinal symptom changes were assessed weekly using the GSRS questionnaire over 4 weeks. The GSRS was subdivided into five domains: dyspepsia (bloating, increased flatulence, belching, borborygmi), abdominal pain (including hunger-induced pain and nausea), reflux (heartburn, acid reflux), constipation, and diarrhea [25]. Each item was scored on a standardized Likert scale: 1–6 for dyspepsia, 1–7 for most symptoms, and 0–3 for stool consistency categories [26].

As shown (Figure 3, Table 2 and Table 3), both groups exhibited symptom improvement after intervention. The Wec120B group showed earlier relief, particularly for dyspepsia and reflux in week 1. The Wec300B group demonstrated more pronounced improvement in weeks 3–4, especially in reflux and constipation. Wec120B was more effective in reducing abdominal pain and diarrhea scores, while Wec300B showed greater improvement in reflux and constipation symptoms. These findings suggest that both formulations are beneficial for GI health, with symptom improvement becoming more pronounced over time.

### 3.4. Modulation of Gut Microbiota Composition

We analyzed the participants’ fecal microbiota using 16S rRNA gene sequencing. Rarefaction curves were used to assess sequencing depth and species richness across samples. The curve for the Wec120B group (Figure 4A) plateaued, indicating adequate sequencing coverage. The Wec300B group (Figure 4B) showed a similar trend with increasing ASV counts stabilizing at higher sequencing depths.

Beta-diversity, which reflects the differences in microbial community composition among groups, was assessed using non-metric multidimensional scaling (NMDS). While the pre- and post-intervention ellipses in both groups showed some overlap, distinct shifts in sample points post-intervention indicated changes in microbial community structure (Figure 4C,D).

Alpha-diversity was evaluated using the Chao1 and ACE indices. A slight decrease in these indices was observed in the Wec120B group, suggesting reduced proportions of low-abundance species and a more optimized microbiota. In contrast, the Wec300B group exhibited increased alpha-diversity post-intervention, reflecting greater microbial richness and diversity (Figure 4E,H).

According to the species annotation results, the top 15 bacterial phyla with the highest abundance at the phylum level in each group were selected to generate a bar graph of species relative abundance. As shown in Figure 5A,C, the dominant taxa included *Firmicutes*, *Bacteroidota*, *Actinobacteriota*, and *Proteobacteria*. Post-intervention, Wec120B increased *Firmicutes*, *Proteobacteria*, and *Actinobacteriota* and reduced *Bacteroidota*. Wec300B increased *Firmicutes* and *Actinobacteriota*, while decreasing both *Bacteroidota* and *Proteobacteria*.

At the genus level (Figure 5B,D), both groups showed post-intervention increases in beneficial genera including *Faecalibacterium*, *Bifidobacterium*, *Blautia*, *Agathobacter*, *Subdoligranulum*, and *Lactobacillus*, and reductions in potentially pathogenic genera such as *Prevotella*, *Escherichia-Shigella*, and *[Ruminococcus]_gnavus_group*. A decrease in *Bacteroides* was observed in both groups, which may result from competition with *Bifidobacterium* for dietary fiber and oligosaccharides [27]. In the Wec300B group, minor genera including *Dialister*, *Parabacteroides*, and *Alistipes* also decreased, possibly due to environmental changes such as pH shifts driven by increased SCFA production [28]. The greater genus-level improvements in the Wec300B group reflect its higher dosage of probiotics and the synergistic action of its multi-strain composition.

Subsequently, statistical analysis was performed using the Wilcoxon algorithm at both the genus and species levels. The post-intervention Wec120B group was compared with the Wec300B group. As shown (Figure 6A,C), significant differences in multiple bacterial genera were observed between the two groups. Specifically, the Wec300B group exhibited significantly higher relative abundances of beneficial bacteria such as *Bifidobacterium*, *Blautia*, *[Eubacterium]_hallii_group*, and *Pediococcus* compared with the Wec120B group, while showing significantly lower relative abundances of pathogenic bacteria including *Prevotella*, *Staphylococcus*, and *Sutterella*. At the species level (Figure 6B,D), significant differences were observed in *Bifidobacterium_longum*, *Blautia_wexlerae, [Eubacterium]_hallii*, *Pediococcus_acidilactici*, *Prevotella_melaninogenica*, *Staphylococcus_warneri*, and *Sutterella_wadsworthensis*. These bacterial species may represent key strains associated with the alleviation of gastrointestinal symptoms

To further investigate the probiotic-induced changes in microbial ecosystem stability and function, we performed Spearman’s correlation analysis to construct genus-level co-occurrence networks before and after the intervention (*p* < 0.05). In these networks, nodes represent genera and edges represent significant correlations (positive or negative). Network density and connectivity reflect ecosystem complexity and stability. Post-intervention (Figure 7B,D), *Bifidobacterium*, *Blautia*, and *Pediococcus* increased in abundance in both groups and showed enhanced correlations with beneficial taxa such as *Lactobacillus*, *Subdoligranulum*, *Fusicatenibacter*, and *Anaerostipes*. These genera also exhibited strong negative correlations with pathogens such as *Prevotella* and *Escherichia-Shigella*. *Bifidobacterium* and *Blautia* became central hubs in the networks, suggesting key roles in microbial interaction and ecosystem functionality. In the Wec300B group, network topology improved significantly, with denser connections and more synergistic relationships involving probiotics, indicating their role as potential “network stabilizers.” The concurrent decrease in pathogen abundance and network marginalization further supports enhanced microbial stability.

To better explore the correlations among gut microbial genera, Spearman’s correlation analysis was performed based on relative abundance data across samples. This analysis identified intra-group relationships between taxa and was used to construct microbial co-occurrence networks. The top 30 genera in total abundance were selected, and pairwise Spearman’s correlation coefficients were calculated to visualize associations among dominant taxa. Only correlations with *p* < 0.05 were considered significant, and the top 30 most significant edges were displayed in the resulting networks.

Following intervention, the Wec120B group showed increased relative abundances of *Bifidobacterium*, *Blautia*, and *[Eubacterium]_hallii_group*, along with stronger negative correlations with *Prevotella*. Additionally, positive correlations were enhanced among beneficial genera including *Bifidobacterium*, *Blautia*, *[Eubacterium]_hallii_group*, *Anaerostipes*, and *Collinsella* (Figure 8A,B). Notably, the correlation between *Bifidobacterium* and *Collinsella*/*Dorea* was strengthened, suggesting that *Bifidobacterium* may have integrated into the symbiotic network and exerted regulatory effects through metabolic cross-feeding, such as lactate production [29]. Furthermore, a negative correlation between *Blautia* and *Escherichia* was observed, indicating that *Blautia* may competitively inhibit pro-inflammatory taxa. In the Wec300B group (Figure 8C,D), *Bifidobacterium*, *Blautia*, and *[Eubacterium]_hallii_group* exhibited stronger negative correlations with *Prevotella*, particularly through the central node *Anaerostipes*. Both *Bifidobacterium* and *Blautia* emerged as key network hubs, forming more cooperative associations with other genera and moving closer to the network core. This highlights their pivotal role in microbial interaction and ecological function, and suggests that they may act as “network stabilizers.” Meanwhile, the decreased abundance and peripheral positioning of certain potential pathogens further support enhanced microbial ecosystem stability following probiotic intervention.

At the species level, the top 30 most abundant taxa were selected for network construction. Only edges with *p* < 0.05 and within the top 30 strongest correlations were retained, and unconnected nodes were excluded. Post-intervention analysis revealed that in the Wec120B group (Figure 9A,B), several core probiotic species—including *Bifidobacterium_animalis*, *Bifidobacterium_longum*, *Collinsella_aerofaciens*, *Bifidobacterium_adolescentis*, and *Bifidobacterium_bifidum*—exhibited enhanced network integration. Specifically, *B. animalis* [30] showed strengthened positive correlations with SCFA-producing species such as *Dialister_sp*. [31], *C. aerofacien* [32], and *Bacteroides_vulgatus* [33], and increased connections with anti-inflammatory species such as *Bacteroides_ovatus* and *Parabacteroides_distasonis* [34]. These findings suggest that *B. animalis* may facilitate cooperative interactions among beneficial bacteria while suppressing potential pathogens.

In the Wec300B group (Figure 9C,D), the probiotic intervention induced more pronounced network reconfiguration. *Bacteroides_ovatus* emerged as a central hub in the post-intervention network, forming highly synergistic relationships with multiple functional species including *B. animalis*, *Adlercreutzia_equolifaciens*, *Bacteroides_uniformis*, *[Eubacterium]_hallii*, *P. distasonis*, *C. aerofaciens*, and *Dialister_sp*. These taxa are known to influence host metabolism, immune regulation, and the coordination of inflammatory responses. Additionally, *B. longum* exhibited enhanced positive correlations with *Anaerostipes_hadrus* [35] and *Bifidobacterium_catenulatum* [36], suggesting strengthened cross-feeding relationships and the increased production of butyrate. Butyrate alleviates intestinal inflammation by inhibiting histone deacetylase (HDAC) activity and modulating G protein-coupled signaling pathways [37]. In the Wec300B group, post-intervention analysis revealed the formation of a tighter and more functionally diverse co-occurrence network, with significantly enhanced integration and cooperative connectivity among core probiotic species. Notably, Wec300B promoted stronger interactions among SCFA-producing bacteria, anti-inflammatory taxa, and mucin-metabolizing microbes. This restructuring of microbial interactions may provide a key microecological basis for the superior efficacy of Wec300B in alleviating gastrointestinal discomfort symptoms.

## 4. Discussion

GI discomfort is a common clinical functional disorder characterized by symptoms such as bloating, abdominal pain, belching, acid reflux, constipation, or diarrhea, which severely affect quality of life. In recent years, the prevalence of GI discomfort has increased due to lifestyle changes [38] and Westernized dietary patterns [39], particularly among young and middle-aged adults. Conventional therapies, including proton pump inhibitors (PPIs), prokinetic agents, and laxatives, primarily target symptom relief but have limited effects on restoring gastrointestinal function. Long-term use may also cause drug dependence and increase safety concerns. Although PPIs are effective, prolonged administration has been linked to higher risks of infections, bone fractures, impaired mineral and vitamin absorption, kidney injury, as well as the disruption of the gastrointestinal barrier and microbiota, warranting a careful evaluation of the risk–benefit ratio [40]. Prokinetic agents may improve gastrointestinal motility, but their clinical benefits are restricted and potential adverse effects, such as movement disorders, arrhythmias, and neurological complications, necessitate the strict control of dosage and treatment duration during long-term use [41]. Increasing evidence has shown that GI dysfunction is closely related to gut microbiota dysbiosis, which may contribute to symptoms via multiple mechanisms, including mucosal barrier disruption and elevated inflammatory mediators [42], impaired gut–brain axis regulation [43], and the reduced production of beneficial metabolites such as short-chain fatty acids (SCFAs), which affect gut motility and immune balance [44]. Therefore, restoring microbial balance and functionality has emerged as a promising strategy for managing GI symptoms, and probiotics have gained attention as a non-pharmacological intervention.

This 4-week randomized, double-blind intervention trial compared two high-potency multi-strain probiotic formulations in alleviating GI discomfort and modulating gut microecology. Symptomatically, the Wec120B group showed faster relief of abdominal pain and dyspepsia, while the Wec300B group demonstrated more sustained improvements in reflux and constipation. Regarding fecal biomarkers, both groups exhibited significant increases in IgA and reductions in FL, FC, NGAL, DAO, D-LA, and LPS, indicating enhanced mucosal immunity, reduced inflammation, and improved gut barrier integrity. Notably, Wec300B achieved greater increases in IgA and reductions in FC, suggesting stronger immunomodulatory effects.

Gut microbial diversity analyses revealed varying degrees of post-intervention changes, indicating the restructuring of the microbial community following probiotic colonization. Community heatmaps and differential taxa bar plots showed increased relative abundances of *Bifidobacterium*, *Blautia*, *[Eubacterium]_hallii_group*, and *Pediococcus*, alongside decreases in *Prevotella*, *Staphylococcus*, and *Sutterella*. *Bifidobacterium* may inhibit pathogen colonization [45], produce SCFAs (e.g., acetate and butyrate) to enhance barrier function and immune regulation [46], and promote anti-inflammatory cytokine secretion [47]. *Blautia* is also strongly associated with host health, particularly in metabolic regulation and immune modulation [48]. Furthermore, GI dysfunction can itself reshape the gut ecosystem via changes in motility [49], gastric acid secretion, and bile acid metabolism [50], altering pH, redox state, and substrate availability, thereby affecting microbial colonization and function. In chronic GI disorders, microbial features typically include reduced diversity, structural imbalance, and an enrichment of pro-inflammatory taxa. For example, beneficial bacteria such as *Bifidobacterium*, *Faecalibacterium*, and *Lactobacillus* are often depleted, while *Escherichia-Shigella*, *Fusobacterium*, and *Clostridium sensu stricto 1* are elevated. Genera like *Ruminococcus* and *Prevotella* also show abnormal trends, suggesting roles in barrier function, immune modulation, and neuroregulation. These features not only underlie GI dysfunction pathophysiology but also represent potential targets for probiotic therapy and microbial diagnostics.

In addition to taxonomic composition, the ecological stability of the gut microbiota determines its resilience and functional potential. Microbial co-occurrence network analysis allows deeper insights into synergistic and antagonistic relationships between genera and structural changes in the microbial ecosystem. Our findings revealed that both Wec120B and Wec300B interventions enhanced microbial network complexity and stability, as reflected by increased node counts, strengthened cooperation among symbiotic genera, and the emergence of key probiotics (*Bifidobacterium*, *Blautia*, *[Eubacterium]_hallii_group*, *Anaerostipes*) as core network members, replacing previously dominant potential pathogens. This structural remodeling may enhance pathogen resistance and improve ecosystem resilience and metabolic capacity.

Spearman’s correlation-based co-occurrence networks further illustrated the differential effects of probiotic activity levels on microbial structure and cooperative function. Both formulations promoted the increased integration and connectivity of core beneficial genera/species, with Wec300B producing stronger effects. This supports the hypothesis that network-level remodeling may underlie the microecological mechanism by which probiotics improve GI symptoms.

At the genus level, the Wec120B group showed increased relative abundances of *Bifidobacterium*, *Blautia*, and *[Eubacterium]_hallii_group* after intervention, with more positive edges observed—particularly with SCFA producers like *Anaerostipes* and *Collinsella*—indicating that low-dose probiotics can support beneficial microbial interactions. The enhanced positive correlations between *Bifidobacterium* and *Dorea*/*Collinsella* may reflect lactate-driven cross-feeding and energy network integration, while the increased negative correlation between *Blautia* and *Escherichia* suggests antagonism via competitive metabolism or inhibitory metabolites. These effects were amplified in the Wec300B group, where *Bifidobacterium* and *Blautia* acted as network hubs forming cooperative structures with *Anaerostipes*, *[Eubacterium]_hallii_group*, and others, organized around the suppression of *Prevotella*. This anti-inflammatory network reflects enhanced microbial homeostasis and suggests that multi-strain probiotics may act as “network stabilizers.”

At the species level, Wec120B intervention enhanced the network integration of several *Bifidobacterium* species *(B. animalis*, *B. longum*, *B. adolescentis*) and *Collinsella aerofaciens*. *B. animalis* displayed stronger positive correlations with SCFA producers such as *Dialister* and *B. vulgatus*, as well as increased links to anti-inflammatory species like *B. ovatus* and *P. distasonis*, indicating its involvement in barrier protection and inflammation control via metabolic coupling. In the Wec300B group, network restructuring was more pronounced. Not only did *B. animalis* maintain its hub status, but *B. ovatus* emerged as a highly connected node, forming a synergistic cluster with *Adlercreutzia equolifaciens*, *B. uniformis*, *[Eubacterium]_hallii*, *P. distasonis*, and others—covering SCFA production, immune regulation, and inflammation control.

Notably, in Wec300B, *B. longum* exhibited stronger positive associations with *Anaerostipes hadrus* and *B. catenulatum*, potentially enhancing butyrate production, which may suppress histone deacetylase (HDAC) activity and modulate G-protein signaling pathways to alleviate inflammation. Additionally, although low-abundance probiotic species such as *Weissella cibaria* did not form stable networks in Wec120B, they showed positive associations with *Akkermansia muciniphila* in Wec300B, suggesting potential cooperation in mucosal barrier metabolism.

In summary, compound probiotic supplementation restructured microbial functionality by reinforcing the synergistic interactions among beneficial taxa, marginalizing potential pathogens, and promoting the formation of anti-inflammatory and energy-related networks. The high-activity formulation Wec300B demonstrated superior effects in enhancing microbial structure and ecological coordination, supporting its mechanistic advantage in the clinical management of GI dysfunction.

Although this study provides valuable evidence regarding the effects of high-activity multi-strain probiotics on gastrointestinal symptoms and gut microbiota, several limitations should be noted.

The absence of a placebo control group represents an important methodological limitation. The randomized, double-blind design ensured a certain degree of rigor, yet without a placebo arm it is difficult to fully exclude potential placebo effects, especially in the context of subjective symptom outcomes such as abdominal pain, bloating, or stool frequency. Placebo responses are well recognized in functional gastrointestinal disorders and may account for a considerable proportion of symptom improvement. Therefore, while our results strongly suggest beneficial effects of probiotic intervention, the causal relationship between treatment and clinical benefit cannot be established with the highest level of certainty. Future trials should incorporate placebo control groups with adequate sample sizes to provide more robust evidence and enhance the reliability of conclusions.

The evaluation of fecal metabolites relied on ELISA-based assays targeting a limited panel of biomarkers. While these markers are clinically relevant and provide practical insights into intestinal immune and barrier function, they do not fully capture the complexity of host–microbiota metabolic interactions. More advanced technologies such as high-resolution mass spectrometry (HRMS) or nuclear magnetic resonance (NMR) could enable untargeted metabolomic profiling, allowing the discovery of novel metabolic signatures linked to probiotic intervention. Moreover, HRMS would facilitate pathway-level analysis and integration with microbiome data, thereby advancing our understanding of the functional consequences of microbial modulation. Given resource and cost constraints, we were unable to employ such techniques in the current trial, but future studies should adopt comprehensive metabolomics to provide mechanistic depth.

In particular, the use of ELISA for measuring gut barrier dysfunction and inflammation-related biomarkers has been well supported in clinical populations. For example, serum D-lactate and diamine oxidase (DAO) levels have been measured by ELISA and shown to be reliable indicators of intestinal permeability in Crohn’s disease patients [51]. Similarly, fecal calprotectin (FC) and lactoferrin (FL) have been extensively assessed using ELISA in clinical cohorts, where FC-ELISA demonstrated a high accuracy in distinguishing inflammatory bowel disease (IBD) from functional disorders [52]. Furthermore, neutrophil gelatinase-associated lipocalin (NGAL) has also been evaluated by ELISA in clinical studies as a biomarker of gut inflammation and barrier integrity [53].

These findings underscore the clinical relevance, robustness, and broad acceptance of ELISA-based biomarker assessment in both gastrointestinal disease and probiotic intervention trials. While HRMS could indeed provide broader metabolomic insights, it was beyond the scope and budget of the present clinical trial.

## 5. Conclusions

In summary, both compound probiotic formulations effectively alleviated GI symptoms in the short term, improved inflammatory (FC, NGAL) and immune (IgA, FL) markers, enhanced intestinal barrier function (LPS, D-LA, DAO), and remodeled the structure and function of the gut microbiota. Compared with Wec120B, the superior performance of Wec300B may be attributed to its higher dosage of probiotics. Key genera such as *Bifidobacterium*, *Blautia*, *Faecalibacterium*, *Agathobacter*, and *Subdoligranulum* may underlie its enhanced effects. These findings offer experimental evidence for the use of compound probiotics in managing GI dysfunction and highlight the importance of strain synergy, dosage, and host-specific factors in shaping probiotic efficacy. Future research should focus on mechanistic studies and personalized intervention strategies to advance the precision application of high-activity probiotics in functional GI disorders.

## Figures and Tables

**Figure 1 nutrients-17-02886-f001:**
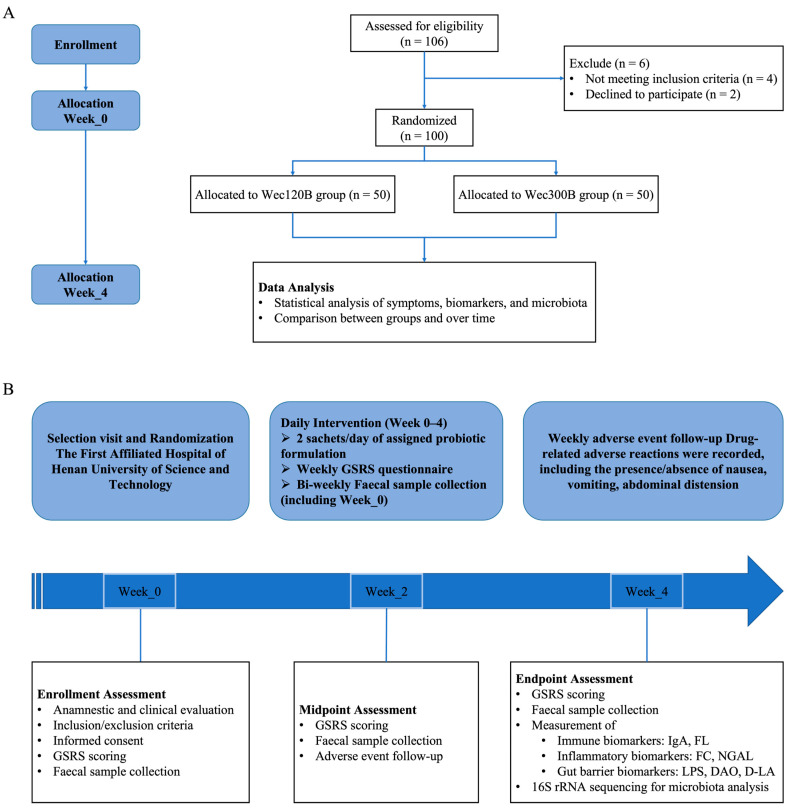
Flowchart for this study. Process of the study implementation. (**A**) The design of the study. (**B**) Flow diagram of this study selection.

**Figure 2 nutrients-17-02886-f002:**
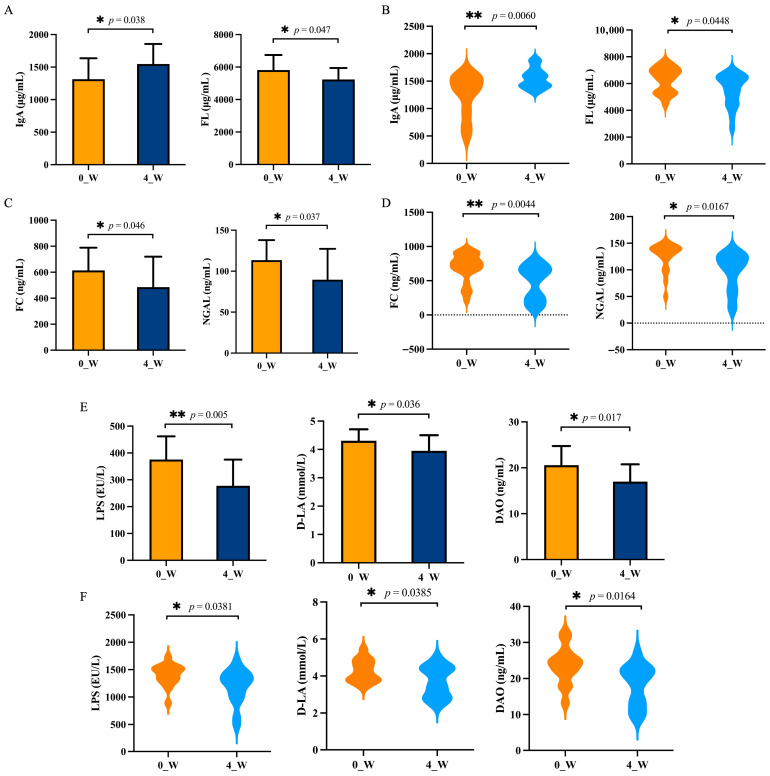
Effects of probiotic intervention on immune factors, inflammatory factors, and intestinal barrier markers. (**A**) Wec120B group immune factors; (**B**) Wec300B group immune factors; (**C**) Wec120B group inflammatory factors; (**D**) Wec300B group inflammatory factors; (**E**) Wec120B group intestinal barrier markers; (**F**) Wec300B group intestinal barrier markers. * *p* < 0.05, ** *p* < 0.01.

**Figure 3 nutrients-17-02886-f003:**
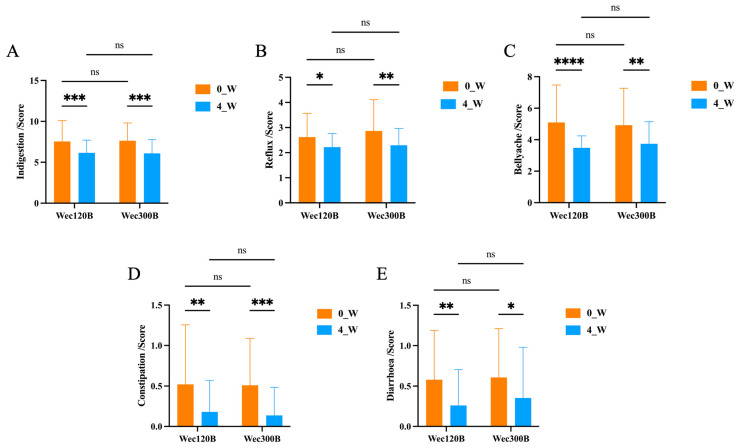
Changes in GSRS domain scores over time. (**A**) Dyspepsia; (**B**) reflux; (**C**) abdominal pain; (**D**) constipation; (**E**) diarrhea. ns *p* ≥ 0.05, * *p* < 0.05, ** *p* < 0.01, *** *p* < 0.001, **** *p* < 0.0001.

**Figure 4 nutrients-17-02886-f004:**
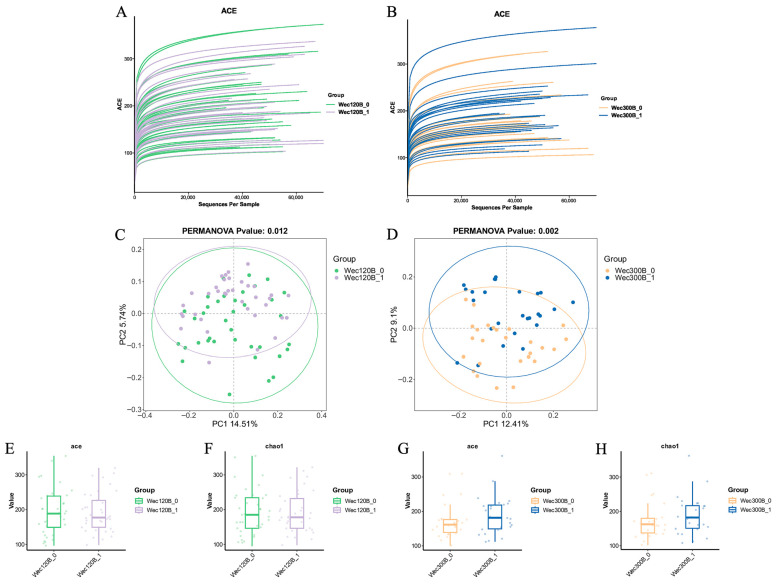
Microbial diversity analysis. (**A**,**B**) Rarefaction curves; (**C**,**D**) NMDS plots; (**E**–**H**) ACE and Chao1 indices.

**Figure 5 nutrients-17-02886-f005:**
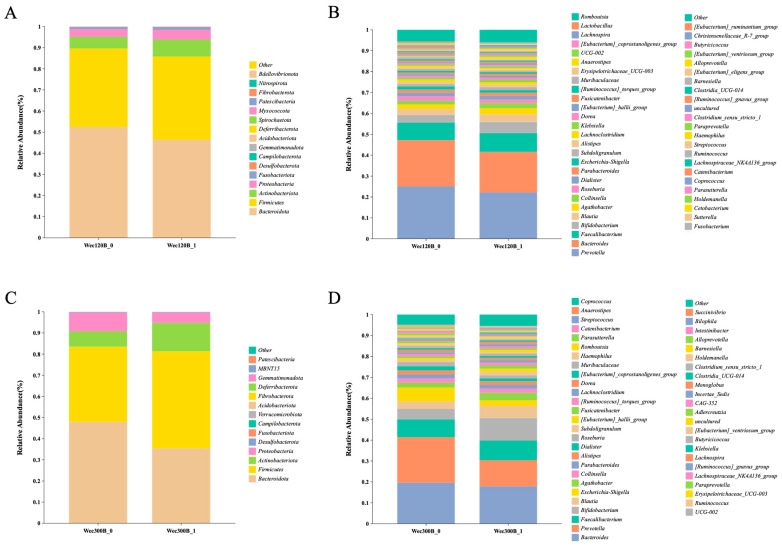
Microbial composition at phylum and genus levels for Wec120B and Wec300B groups. (**A**,**C**) Relative abundance of bacterial communities at the phylum level; (**B**,**D**) relative abundance of bacterial communities at the genus level.

**Figure 6 nutrients-17-02886-f006:**
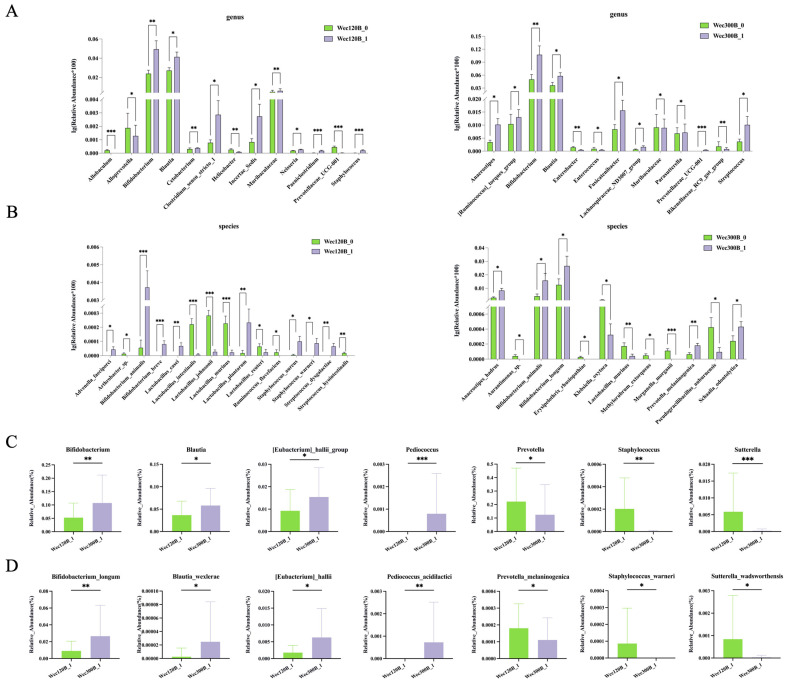
Significant differences in intestinal microbiota at the genus and species levels. (**A**) Species with genus-level differences in the Wec120B group and Wec300B group before and after the intervention; (**B**) species-level differences between Wec120B group and Wec300B group before and after intervention; (**C**) species with genus-level differences between the Wec120B group and the Wec300B group after the intervention; (**D**) species-level differences between the Wec120B group and the Wec300B group after intervention. * *p* < 0.05, ** *p* < 0.01, *** *p* < 0.001.

**Figure 7 nutrients-17-02886-f007:**
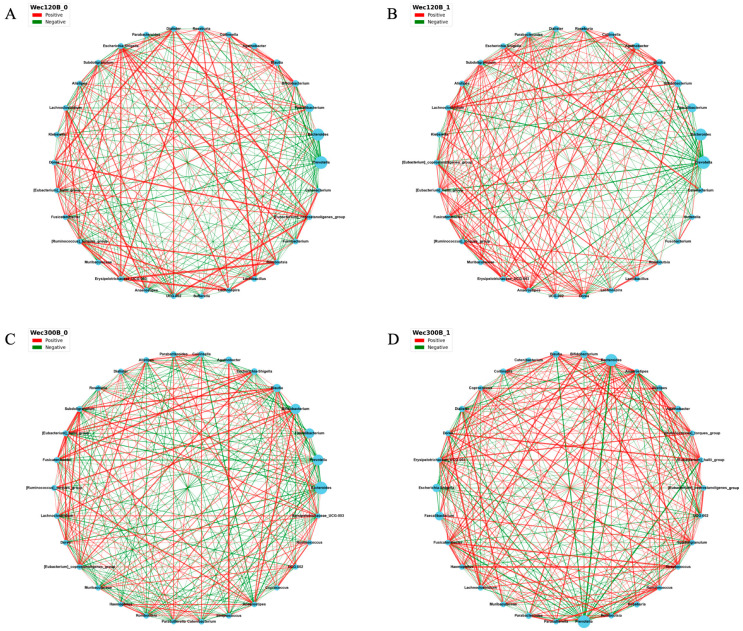
Microbial community interaction network analysis. (**A**) Wec120B group before intervention; (**B**) Wec120B group after intervention; (**C**) Wec300B group before intervention; (**D**) Wec300B group after intervention.

**Figure 8 nutrients-17-02886-f008:**
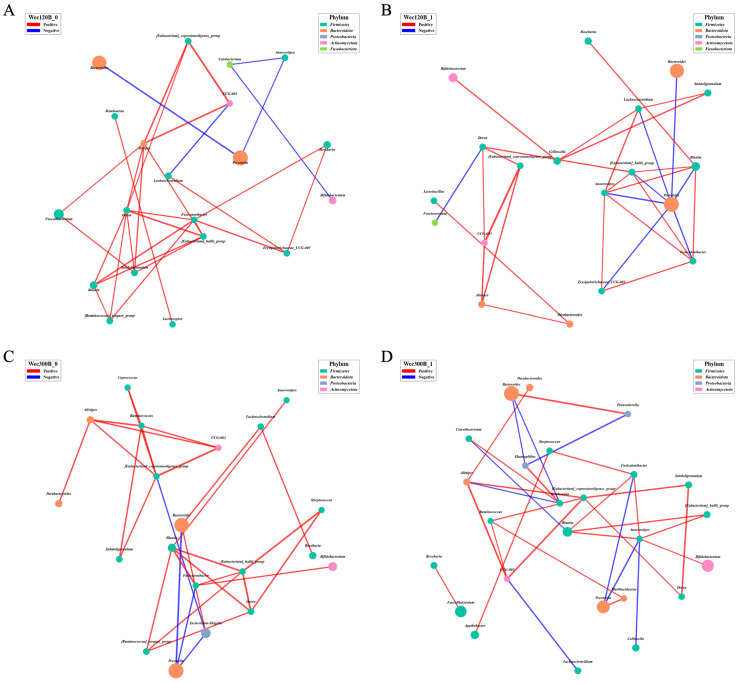
Genus-level microbiome correlation network. (**A**) Wec120B group before intervention; (**B**) Wec120B group after intervention; (**C**) Wec300B group before intervention; (**D**) Wec300B group after intervention.

**Figure 9 nutrients-17-02886-f009:**
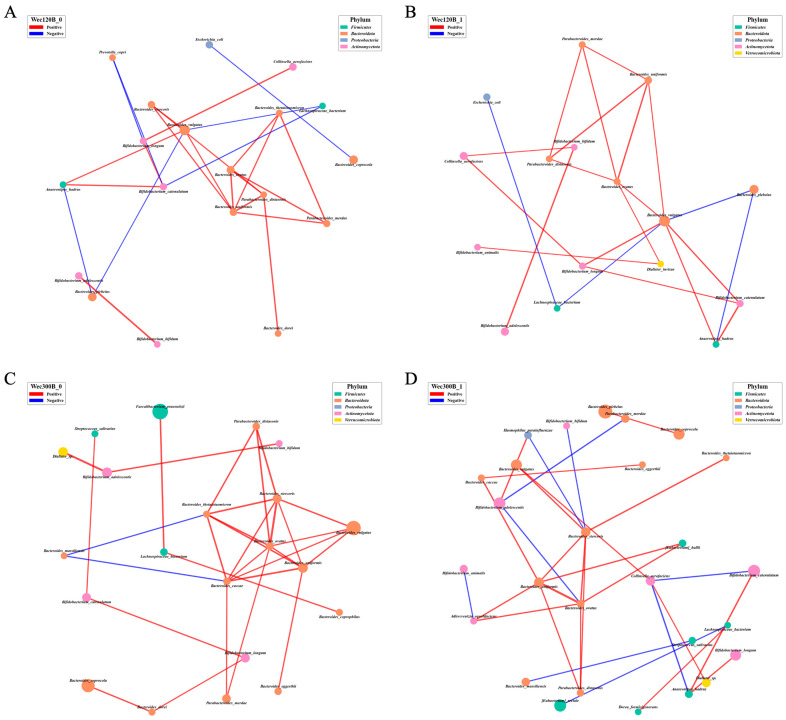
Species-level microbiome correlation network. (**A**) Wec120B group before intervention; (**B**) Wec120B group after intervention; (**C**) Wec300B group before intervention; (**D**) Wec300B group after intervention.

**Table 1 nutrients-17-02886-t001:** Baseline characteristics of study participants.

Group	Age ($\bar{x}\pm s$)	Gender	
		Male	Female
Wec120B group (*n* = 50)	21.1 ± 5.18	29 (58%)	21 (42%)
Wec300B group (*n* = 50)	20.42 ± 2.7	24 (48%)	26 (52%)

**Table 2 nutrients-17-02886-t002:** Analysis of GSRS in Wec120B group.

Domain	Evaluation	Score	*p*-Value (vs. Baseline)	95% CI (vs. Baseline)
Indigestion	Baseline	7.54 ± 2.56	-	-
	W1	6.64 ± 2.10	0.057	−0.02815, 1.82815
	W2	6.44 ± 1.79	0.014	0.22461, 1.97539
	W3	6.52 ± 1.58	0.018	0.17625, 1.86375
	W4	6.16 ± 1.54	0.0008	0.57955, 2.18025
Reflux	Baseline	2.74 ± 1.38	-	-
	W1	2.44 ± 0.88	0.199	−0.16052, 0.76052
	W2	2.62 ± 1.05	0.626	−0.36675, 0.60675
	W3	2.26 ± 0.60	0.027	0.05715, 0.90285
	W4	2.22 ± 0.55	0.015	0.10296, 0.93704
Bellyache	Baseline	5.10 ± 2.38	-	-
	W1	4.24 ± 1.52	0.033	0.06863, 1.65137
	W2	4.10 ± 1.40	0.012	0.22569, 1.77431
	W3	3.58 ± 1.00	0.000	0.79758, 2.24242
	W4	3.48 ± 0.76	<0.0001	0.91984, 2.32016
Constipation	Baseline	0.52 ± 0.58	-	-
	W1	0.38 ± 0.60	0.239	−0.09467, 0.37467
	W2	0.36 ± 0.48	0.138	−0.05215, 0.37215
	W3	0.32 ± 0.59	0.090	−0.03157, 0.43157
	W4	0.26 ± 0.44	0.013	0.05518, 0.46482
Diarrhea	Baseline	0.54 ± 0.73	-	-
	W1	0.38 ± 0.64	0.247	−0.11251, 0.43251
	W2	0.34 ± 0.66	0.155	−0.07672, 0.47672
	W3	0.30 ± 0.54	0.066	−0.01647, 0.49647
	W4	0.18 ± 0.39	0.003	0.12691, 0.59309

**Table 3 nutrients-17-02886-t003:** Analysis of GSRS in Wec120B group.

Domain	Evaluation	Score	*p*-Value (vs. Baseline)	95% CI (vs. Baseline)
Indigestion	Baseline	7.63 ± 2.18	-	-
	W1	6.73 ± 2.05	0.034	0.07031, 1.73361
	W2	6.73 ± 2.15	0.038	0.05014, 1.75379
	W3	6.53 ± 2.00	0.009	0.27520, 1.92088
	W4	6.10 ± 1.68	0.0002	0.73682, 2.32270
Reflux	Baseline	2.94 ± 1.33	-	-
	W1	2.57 ± 0.92	0.104	−0.07770, 0.82280
	W2	2.61 ± 1.08	0.168	−0.14298, 0.80965
	W3	2.27 ± 0.57	0.001	0.26412, 1.06922
	W4	2.29 ± 0.67	0.003	0.23236, 1.06176
Bellyache	Baseline	4.92 ± 2.35	-	-
	W1	4.20 ± 2.00	0.096	−0.13144, 1.58242
	W2	4.27 ± 1.67	0.112	−0.15412, 1.44824
	W3	3.98 ± 1.61	0.020	0.15084, 1.73152
	W4	3.75 ± 1.40	0.003	0.41730, 1.93564
Diarrhea	Baseline	0.51 ± 0.61	-	-
	W1	0.49 ± 0.67	0.878	−0.23346, 0.27268
	W2	0.38 ± 0.60	0.255	−0.10065, 0.37516
	W3	0.31 ± 0.58	0.101	−0.03875, 0.43091
	W4	0.35 ± 0.63	0.039	0.01334, 0.49647
Constipation	Baseline	0.51 ± 0.58	-	-
	W1	0.55 ± 0.73	0.764	−0.29796, 0.21953
	W2	0.32 ± 0.65	0.110	−0.04524, 0.43739
	W3	0.22 ± 0.54	0.009	0.07406, 0.51418
	W4	0.14 ± 0.35	0.000	0.18501, 0.56008

## Data Availability

All original contributions presented in the study are included in the article; further inquiries can be directed to the corresponding author.

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
