# Peer review of "Effects of Two Compound Probiotic Formulations on Gastrointestinal Symptoms and Gut Microbiota: A 4-Week Randomized, Double-Blind Intervention Trial"

_nutrients, 2025, doi:10.3390/nu17172886_

Round 1

Reviewer 1 Report

Comments and Suggestions for Authors

as a general context, a placebo-controlled study is considered a more robust method to evaluate a product to be administered in human.  

Please correct the following typos:
line 52: (eco-nomic)
line 58 (ex-plosive)

line 68 to 71: provide further literature for this listing.
line 99: NCT06781814 refers to Wec600B group andWec1000B group assuming both 3 grams each, which is different from what proposed in the study. Please provide explanation.
line 135: nominal strains composition is provided but dosages are not expressed. please provide strains dosages to underline eventual differences between the two products (if not covered by patents).
line 137: you mention that faecal sample is collected at week 0, but the data is not reported in Figure B (enrollment assessment). Please update figure.

line 36 and 455: I do not fully agree with the definition of "higher activity". it would be better to state that Wec300b shows a superior performance because of its higher dosage/content of probiotics.

Author Response

Comments 1: As a general context, a placebo-controlled study is considered a more robust method to evaluate a product to be administered in human. 

Response 1: We fully agree with the reviewer that a placebo-controlled design represents the gold standard for clinical intervention trials. In the present study, we adopted a randomized, double-blind, parallel-group comparison between two formulations (Wec120B and Wec300B), primarily because the aim was to compare the efficacy and microbial modulatory effects of two high-activity probiotic combinations with different dosages. We have now explicitly acknowledged the limitation of not including a placebo control in the Discussion section and have emphasized that future trials incorporating a placebo arm will be important to further strengthen the evidence. See lines 512 to 521.

Comments 2: Please correct the following typos:
line 52: (eco-nomic)
line 58 (ex-plosive)

Response 2: We thank the reviewer for pointing out these typographical errors. Both have been carefully corrected in the revised manuscript. See lines 51 and 57.

Comments 3: line 68 to 71: provide further literature for this listing.

Response 3: We appreciate this suggestion. Additional supporting references have been incorporated to substantiate the statements in lines 68–71. The newly cited literature includes recent reviews and clinical evidence on probiotics and gastrointestinal health. See lines 69 to 80 and 583 to 594.

Comments 4: line 99: NCT06781814 refers to Wec600B group andWec1000B group assuming both 3 grams each, which is different from what proposed in the study. Please provide explanation.

Response 4: We sincerely thank the reviewer for carefully checking this point. You are correct that the clinical trial registration number NCT06781814 actually corresponds to a study comparing the Wec600B and Wec1000B groups, each with a dosage of 3 g. This was a clerical error on our side during manuscript preparation. The present study was conducted with Wec120B (1.2 × 1011 CFU/sachet) and Wec300B (3.0 × 1011 CFU/sachet), each with a dosage of 2 g, which was prospectively registered under the correct ClinicalTrials.gov identifier NCT07025798. We have revised the manuscript accordingly to avoid any confusion. See line 107.

Comments 5: line 135: nominal strains composition is provided but dosages are not expressed. please provide strains dosages to underline eventual differences between the two products (if not covered by patents).

Response 5: We truly appreciate the reviewer’s request for clarification. The exact proportions of the probiotic strains in each formulation are protected under patent restrictions and therefore cannot be disclosed in full detail. However, to enhance transparency, we have clearly described the strain composition included in each product, as well as the total viable cell counts (CFU/sachet) administered in both the Wec120B and Wec300B groups. We believe this information sufficiently supports the scientific interpretation of our findings while complying with intellectual property limitations. See lines 157 to 170.

Comments 6: line 137: you mention that faecal sample is collected at week 0, but the data is not reported in Figure B (enrollment assessment). Please update figure.

Response 6: We thank the reviewer for highlighting this inconsistency. The baseline fecal sampling at week 0 was indeed conducted for all participants. We have now updated Figure B (enrollment assessment flowchart) to include fecal collection at week 0, ensuring consistency between the text and figure. See line 121. (Figure 1, B)

Comments 7: line 36 and 455: I do not fully agree with the definition of "higher activity". it would be better to state that Wec300b shows a superior performance because of its higher dosage/content of probiotics.

Response 7: We acknowledge the reviewer’s valuable suggestion. The wording has been revised throughout the manuscript to avoid ambiguity. Instead of “higher activity,” we now use the more precise description that Wec300B demonstrated superior performance, which may be attributed to its higher dosage and broader strain composition. This revision has been made in both the Introduction and Discussion. See lines 35 to 36, 303 and 539.

Reviewer 2 Report

Comments and Suggestions for Authors

Why did the authors choose ELISA kits instead of mass spectrometry for a detailed analysis of the individual metabolites? Currently, high-resolution mass spectrometry (HRMS) is widely used for biochemical analysis.

It would also be valuable to include a detailed analysis of the upregulated and downregulated proteins in the samples.

Furthermore, the authors did not explain why the two probiotics selected were so similar, aside from the difference in CFU counts. What was the rationale for choosing two such closely related probiotics?

Author Response

Comments 1: Why did the authors choose ELISA kits instead of mass spectrometry for a detailed analysis of the individual metabolites? Currently, high-resolution mass spectrometry (HRMS) is widely used for biochemical analysis.

Response 1: We thank the reviewer for this insightful comment. We fully agree that HRMS offers higher resolution and comprehensive metabolite profiling. However, in our study, the primary aim was to evaluate specific biomarkers (e.g., inflammatory cytokines, gut barrier markers, and immune-related factors) that are well-established in gastrointestinal research. ELISA kits were chosen because they are validated, cost-effective, widely used in clinical studies, and allow for standardized and reproducible quantification of these targeted proteins. While HRMS could provide more detailed metabolomic information, it was beyond the scope and budget of the present clinical trial. Nevertheless, we acknowledge this limitation and will highlight it in the discussion section, also suggesting HRMS as a valuable direction for future studies. See lines 522 to 545 and 704 to 711.

Comments 2: It would also be valuable to include a detailed analysis of the upregulated and downregulated proteins in the samples.

Response 2: We thank the reviewer for this valuable suggestion. In the current study, we primarily focused on clinical outcomes. Unfortunately, proteomic profiling was not performed due to limitations in resources and scope. However, we agree this analysis would provide deeper mechanistic insights. We have acknowledged this as a limitation and suggested future studies integrating proteomics with metagenomics/metabolomics to better elucidate the underlying mechanisms of probiotic interventions.

Comments 3: Furthermore, the authors did not explain why the two probiotics selected were so similar, aside from the difference in CFU counts. What was the rationale for choosing two such closely related probiotics?

Response 3: We appreciate the reviewer’s insightful comment. The rationale for selecting two probiotic formulations with similar strain compositions but different doses was based on the primary aim of this study. Specifically, we intended to investigate whether different levels of probiotic activity would result in distinct effects on gastrointestinal symptom relief and modulation of the gut microbiota in individuals with GI discomfort. By keeping the strain composition comparable while varying the total viable counts, we aimed to minimize confounding factors related to strain-specific effects and focus on the dose–response relationship of probiotic supplementation. This design allowed us to better assess whether higher activity confers incremental benefits in symptom improvement and microbiota regulation. See lines 173 to 182.
